# DFMG decreases angiogenesis to uphold plaque stability by inhibiting the TLR4/VEGF pathway in mice

**Pingjuan Bai** [1,2], **Xueping Xiang**[2], **Jiawen Kang**[2], **Xiaoqing Xiang**[2], **Jingwen Jiang**[2], **Xiaohua Fu**[2], **Yong Zhang**[2]*, **Lesai Li**[3]*

**1** Pathology Department, Jiangxi Maternal and Child Health Hospital, Nanchang, Jiangxi Province, China, **2** Medical College of Hunan Normal University, Changsha, Hunan Province, China, **3** Hunan Cancer Hospital, Changsha, Hunan Province, China

* Thaddeus@hunnu.edu.cn (YZ); LLS0731@126.com (LL)

**Data Availability Statement:** All relevant data are within the manuscript and its Supporting Information files.

## Abstract

The aim of this study was to elucidate the specific mechanism through which 7-difluoro-methoxy-5,4'-dimethoxygenistein (DFMG) inhibits angiogenesis in atherosclerosis (AS) plaques, given its previously observed but poorly understood inhibitory effects. In vitro, a model using Human Umbilical Vein Endothelial (HUVEC-12) cells simulated the initial lesion in the atherosclerotic pathological process, specifically oxidative stress injury, by exposing cells to 30 μmol/L LPC. Additionally, an AS mouse model was developed in ApoE knockout mice through a 16-week period of high-fat feeding. DFMG demonstrated a reduction in tubule quantities in the tube formation assay and neovascularization induced by oxidative stress-damaged endothelial cells in the chicken embryo chorioallantoic membrane assay. Furthermore, DFMG decreased lipid levels in the blood of ApoE knockout mice with AS, along with a decrease in atherosclerotic plaques and neovascularizations in the aortic arch and descending aorta of AS animal models. DFMG treatment upregulated microRNA140 (miR-140) expression and suppressed VEGF secretion in HUVEC-12 cells. These effects were counteracted by Toll-like receptor 4 (TLR4) overexpression in HUVEC-12 cells subjected to oxidative injury or in a mouse model of AS. Dual-luciferase reporter assays demonstrated that miR-140 directly targeted TLR4. Immunohistochemical assay findings indicated a significant inverse relationship between miR-140 expression and TLR4 expression in ApoE knockout mice subjected to a high-fat diet. The study observed a close association between DFMG inhibitory effects on angiogenesis and plaque stability in AS, and the inhibition of the TLR4/NF-κB/VEGF signaling pathway, negatively regulated by miR-140.

## Introduction

Atherosclerosis (AS) is a common pathological lesion in the arteries and its associated complications such as cardiovascular disease (CVD) remains one of the leading causes of death and disability worldwide [1, 2]. Acute cardiovascular events, such as myocardial infarction, stroke,

**Funding:** This study is supported by the Natural Science Foundation of China (Grant No. 81370382) and the Natural Science Foundation of Hunan Province (Grant No. 14JJ2059 and 2022JJ30415). Xiaohua Fu and Yong Zhang, as awardees, have provided valuable reviews for this article.

**Competing interests:** The authors have declared that no competing interests exist.

etc., are the most serious complications of AS, which are caused by plaque instability and vulnerability during the development of AS. Over 17 million people died from CVD in 2015, representing 31% of all global deaths [3, 4]. At present, lipid-lowering pharmacological therapy, non-lipid-lowering therapy, and intravenous thrombolysis are the main treatments for AS [5], which can relieve clinical symptoms, but have poor long-term effects in the patients with AS. Nevertheless, due to the ubiquitous tendency of AS development in human beings, there remains a need to find new ways to understand and treat AS.

Angiogenesis, the formation of new blood vessels from the endothelium, in atherosclerotic lesions plays a critical role in growth and instability of plaques [6, 7]. Atherosclerotic plaque rupture and symptomatic coronary heart disease are strongly associated with the presence of intraplaque and adventitial angiogenesis. There is evidence that angiogenesis occurs in the early stages of AS, and the density of vasa vasorum in AS-prone areas is higher than in other parts [8, 9]. In addition, the new-born blood vessels in the plaque have high permeability, releasing red blood cells from the plaque to cause bleeding, and it is found that the angiogenesis in the plaque is closely related to the thin cap plaque and the vulnerable plaque. Endothelial dysfunction initiated the progress of AS and played a central role in many vascular-related disorder [10, 11]. Studies have suggested that in the foci of atherosclerotic plaques, reducing angiogenesis may protect the formation of atherosclerotic unstable plaques or prevent the rupture of vulnerable plaques [12, 13]. Koutouzis et al found that statins can inhibit angiogenesis in AS plaques and stabilize the plaques in morphology by observing human carotid endarterectomy specimens [14]. Bevacizumab, a widely recognized anti-angiogenic drug, can effectively reduce the size of atherosclerotic plaques and improve plaque development [15]. However, the side effects of anti-angiogenic therapy for AS often outweigh the therapeutic effect, increasing the risk of cardiovascular and cerebrovascular diseases. Inhibiting immature neovascularization while maintaining an appropriate level of mature neovascularization may benefit plaque stabilization and even lipid transfer [16, 17]. Unfortunately, the discovery of new drugs that inhibit intraplaque angiogenesis has been slow. Therefore, further studies are needed to discover new antiangiogenic drugs that with acceptable side effects and to evaluate their potential efficacy in preventing plaque instability in AS.

7-difluoromethoxy-5,4-dimethoxygenistein (DFMG) is a new chemical entity synthesized with genistein as the leading compound by our team, and previously found it played a role in preventing the occurrence of AS [18, 19]. Recently, we found that DFMG also had the effect of maintaining plaque stability in mice with AS, which may be related to DFMG reducing angiogenesis at the injury site. This study mainly explored the mechanism of DFMG on maintaining plaque stability in animals with developed AS.

## Materials and methods

### Reagents

DFMG (purity>99%, self-synthesized, C18H14O5F2, 348(kDa). Lysophosphatidylcholine (LPC) was purchased from Sigma-Aldrich (St Louis, USA). DMSO, HE Staining Kit and Masson's Trichrome Stain Kit were provided by Solarbio (Beijing, china). Toll-like receptor 4 (TLR4), Nuclear factor κB (NF-κB), Vascular endothelial growth factor (VEGF), Vascular endothelial growth factor 2(VEGFR2) and von Willebrand Factor (vWF) antibodies were purchased from Biosciences (OH, USA). Matrigel was purchased from BD Biosciences; Cell Counting Kit-8(CCK8) and lactate dehydrogenase (LDH) kits were purchased from Nanjing Jiancheng Biotechnology Co., Ltd. has-miR-140-5p-mimic, hsa-miR-140-5p-inhibitor and TLR4-overexpression eukaryotic plasmid were purchased from Shanghai Genechem Co.,Ltd. TLR4-knock down eukaryotic plasmid was purchased from Hunan Aijia biology co., Ltd.

## Mice

ApoE$^{-/-}$ mice, TLR4$^{-/-}$ mice and C57BL/6 mice with the same genetic background were donated by the Institute of Model Animal, Wuhan University (Animal Qualification Test Report No.: BJYRL-WBKH-20160224A1). ApoE$^{-/-}$ mice and TLR4$^{-/-}$ mice were screened by DNA, and one male and two female mice were randomly selected to mate in a cage. Six-week-old double gene knockout (ApoE$^{-/-}$/TLR4$^{-/-}$) mice, ApoE$^{-/-}$ mice and ApoE$^{-/-}$/TLR4$^{-/-}$ mice were chosen for experimental study with 10 mice in each group. Mice were fed with high-fat diet (purchased from Hunan Shrek Jingda Animal Company, containing 10% lard, 10% egg yolk powder, 2% cholesterol, 0.2% cholic acid) for 16 weeks. ApoE$^{-/-}$ + DFMG group and ApoE$^{-/-}$/TLR4$^{-/-}$ + DFMG group was fed with high fat and DFMG (10mg/) kg·d)) was added for intervention. C57BL/6 mice were served as Wild Type group. Wild Type group was fed with routine diet without any drug intervention. After 16 weeks, mice were weighed and sacrificed, paraffin sections of aorta were prepared for some morphological staining. Flowchart of animal experiment is shown in S1 Fig. All experimental mice were housed in the SPF animal room of the Medical College of Hunan Normal University, maintaining a 12-hour / 12-hour (light / dark) environment every day. All procedures involving animals were performed in strict accordance with ARRIVE guidelines for the care and use of laboratory animals, and were approved by Ethics Committee of Hunan Normal University (No.2013289).

## Cell culture and treatment

Human umbilical vein endothelial (HUVE-12) cells in logarithmic growth phase were cultured in Roswell Park Memorial Institute-1640 (RPMI-1640) medium containing 10% fetal bovine serum, 1% penicillin and streptomycin and cultured in 37˚C incubator with 5%CO2 concentration and fully humidified atmosphere. After being digested with 0.25% trypsin, the single cell suspension was counted and planted in the pore plate. The method of constructing endothelial cell model of oxidative stress-induced injury is as described in previous studies [20]. The cells were seeded in 6-well plates with a cell density of $5 \times 10^5$/mL. The lentivirus mixture containing recombinant (serum-free 1mL containing 5ug/mL polybrene with a titer of $1 \times 10^8$TU/mL lentivirus 5uL) was incubated for 48 hours, and the cells were cultured and screened in a complete medium containing puromycin (2.5ug/mL). The fluorescence of the cells was observed under an inverted fluorescence microscope, and the fluorescence of the cells was stably maintained at more than 95%. TLR4 and miR-140 expression was detected by Real Time Polymerase Chain Reaction (RT-qPCR) or Western blot (WB) analysis. The structure of the plasmid vector is shown in S2 Fig. The cell groups were categorized as follows: NC group, DFMG group, TLR4$^{-/-}$ group, TLR4$^{-/-}$+DFMG group, TLR4$^{+/+}$ group, TLR4$^{+/+}$ +DFMG group, miR-140$^{-/-}$ group, miR-140$^{-/-}$+DFMG group, miR-140$^{+/+}$ group, and miR-140$^{+/+}$ +DFMG group. Except for the NC group, all other groups underwent pretreatment with 30μM LPC, employing the same pretreatment method as previously described [19].

## Chicken chorioallantoic membrane (CAM) experiment to evaluate angiogenesis

White-skinned eggs (Lujian ecological breeding farm, Hunan) were placed head-up for several hours or overnight, then washed in 37˚C water supplemented with bromogeramine and incubated in an incubator (Liming Incubation Equipment, Guangzhou). On the third day of incubation, the eggs without blood filaments were selected for the following experiments. Gelatin sponge cubes cut into 5mm were immersed in cell supernatants of each group and placed on the CAM of an artificial air chamber on the 10th day of egg incubation. Finally, the air

chamber was sealed with sealing membrane and incubated for 72 hours. The vascular growth of CAM membrane was observed under somatotype microscope. The blood vessels of chicken CAM were photographed and counted by ImageJ [ImageJ is a java-based data analysis software developed by the National Institutes of Health (NIH)].

## Tubule formation assay

The Matrigel, microcentrifuge tubes, pipette tips, and 24-well plate were precooled in a refrigerator at 4°C for several hours. The Matrigel was diluted with RPMI-1640 basic medium at 1:3. 200μl of Matrigel was added to each well (concentration > 10 mg/ml; BD Matrigel™ 354230) and allowed to solidify at room temperature for 30 minutes. HUVE-12 cells in logarithmic growth phase were digested into single cell suspension with 0.25% trypsin and mixed with the supernatant of cells from each group. The cells were spread on the Matrigel with a cell density of $5 \times 10^5$ cells per well. After incubation for 8 hours, the formation of cell tubes was observed under microscope, and the area of cell tube formation was measured using ImageJ.

## Cell proliferation assay and LDH assay

The cells in logarithmic growth phase were seeded in 96-well plate at the density of $4 \times 10^4$. The LPC group was treated with 30% LPC for 24 hours. The DFMG+LPC group was pretreated with 3.0μM concentration of DFMG for 2 hours and then added with 30μM concentration of LPC for 24 hours. then added with the CCK-8 solution of 100μL to each well, and incubated for one more hour. The absorbance at 450nm was measured by enzyme labeling instrument (BIO-TEK, America) to observe the cell proliferation activity. Cell damage was detected by the LDH reagent, and the methods were performed according to the kit instructions. The absorbance at 450nm was measured by enzyme labeling instrument, and the final data were calculated according to the formula of instructions.

## Immunofluorescence

HUVE-12 cells in logarithmic growth phase are grown in 12-well plates. Absorb 300μL 4% paraformaldehyde in the climbing hole and place it for 15 minutes to make the cells fixed. Add 300μL of 0.5%TritonX-100 (prepared by PBS) to each hole, and place 20min at room temperature to make the cell membrane permeable. The specific antigen was blocked by goat serum and 30min was placed at room temperature. Put the cell climbing slide in a wet box, add a sufficient amount of primary antibody and spend the night in a 4°C refrigerator; the next day, incubate 20min at room temperature, add diluted fluorescent secondary antibody to the climbing tablet, avoid light and incubate at 20–37°C for 1 hour, then wash the cell climbing slide with PBS-T for 3 times, each time 3min. Finally, DAPI (from Vector, Germany) was used for re-staining and nail polish sealing, and then the images were observed and collected under fluorescence microscope.

## Identification of mouse genotype

The agarose gel with a concentration of 0.75% was prepared with double distilled water, and the agarose solution was poured into the gel mold and placed at room temperature for 30 minutes to completely solidify the gel. Take out the DNA sample and add $5 \times$ sample buffer in proportion and mix it well. The conditions of electrophoresis are as follows: the voltage is adjusted to 110V, the current is set at 45mA, and the electrophoresis is stopped when the band moves to about 2cm from the front of the gel. At the end of electrophoresis, take out the gel plate to develop and take pictures.

### Histological and immunohistochemical studies

The formalin-fixed aorta was dehydrated in ethanol and embedded in paraffin, and 5μm serial sections were stained with hematoxylin and eosin (Millipore-Sigma). Sections were evaluated blindly to score for pathological changes. The expression of TLR4, VEGF, VEGFR2, vWF and NF-κB were examined by immunohistochemical staining with individual antibodies with appropriate controls suggested by manufactures. The stained tissue sections were scanned, positive areas outlined by 2 board-certified pathologists (blind for the animal treatment conditions), and positive area quantified by Image-J software.

### Evaluation of lipid infiltration in atherosclerotic plaques

AS in the aortic arch, descending thoracic aorta, and abdominal aorta was assessed by fastidiously removing perivascular adipose tissue from the excised aorta, longitudinally incising it, staining it with Oil Red O (Solarbio, China), and performing en face quantification of total plaque area with computerized software. To measure the en face lesion area of the aorta, mice were euthanized and then perfused with 1X PBS with a constant pressure via the left ventricle. The whole aorta was immediately removed, fixed with 10% neutral buffered formalin for 5 minutes, then the aorta was washed with PBS, and stained with Sudan IV solution for 30 minutes. Excess stain was washed off with 70% ethanol for 30 minutes. The aorta was cut open longitudinally following carefully dissecting out adipose tissue in adventitia. The opened aorta was spread on a clean glass slide, followed by covering with another glass slide on the top, and then injecting water inside between 2 slides, followed by glue the edge of slides. The luminal side of the stained aorta was photographed. Adobe Photoshop and Image-J (Media Cybernetics, Rockville, MD) were used to perform the image capture and analysis. The extent of AS was expressed as the percent of surface area of the entire aorta covered by lesions.

### The content of collagen fibers in AS plaques was detected by Masoon staining

The tissue sections of mouse thoracic aorta were baked, dewaxed and hydrated, and hematoxylin staining solution was added to the tissue for nuclear staining; 1% hydrochloric acid solution was used for differentiation; then Masson compound staining solution was added to the tissue and rinsed with distilled water; 1% phosphotungstic acid solution was added to the tissue, and then aniline blue solution was added to re-dye 5min, and finally 1% glacial acetic acid was used to rinse tissue 1min. Xylene is transparent and sealed with neutral gum. The results showed that the collagen fibers were blue and the cytoplasm, muscle fibers and red blood cells were red. The sections were scanned and analyzed by automatic quantitative pathological imaging system, and the content of collagen fibers in tissue was quantitatively analyzed by Inform software [21].

### Detection of blood lipids

The mice were subjected to a 12-hour fasting period with water deprivation before the experiment. Subsequently, anesthesia was induced through abdominal cavity access for blood collection from the orbital cavity. The collected blood samples were left at room temperature for 1 hour to promote clotting before centrifugation. Serum lipid levels in mice were then analyzed and observed using an automatic biochemical analyzer.

### Western blot

SDS lysis buffer was used to extract tissue or cell proteins, and total protein concentration was measured using a bicinchoninic acid protein assay kit (Beyotime Biotechnology, China).

Protein was separated by 4–20% Tris-Glycine SDS-PAGE gel before transfer to a PVDF membrane. Equal protein concentrations were loaded and electrophoresed on 4% to 12% gradient gels (Bio-Rad, United Kingdom) and transferred to 0.2-μm nitrocellulose membranes. Blots were blocked with 5% (w/v) skimmed milk powder and incubated overnight at 4°C with TLR4, NF-κB or VEGF antibody diluted in Signal Boost Solution 1 (Merck Millipore). Optical density of bands was quantified using Gel Doc XR+ Gel Documentation System from Bio-Rad and normalized to sample total protein content (whole lane) assessed using Bio-Rad stain-free technology and present the most intense band within the representative figures, which are referred to as loading control.

## Quantitative real-time PCR

RNA was extracted with Trizol (Vazyme Biotech Co.,Ltd, China) from cells or aorta of mice. For quantitative real-time polymerase chain reaction, total RNA was converted to cDNA using random primers, reverse transcribed using SuperScript II (Invitrogen), and analyzed using SYBR Green assay with specific primers from Applied Biosystems with specific primers. The primer sequence of TLR4 was (TLR4-F: 5′-CCGAGGCCATTATGCTATGT-3′, TLR4-R: 5′-TCCCTTCCTCCTTTTCCCTA-3′), VEGF primer sequence (VEGF-F: 5′-GCACATA GAGAGAATGAGCTTCC-3′; VEGF-R: 5′-CTCCGCTCTGAACAAGGCT-3′); miR-140 primer sequence (miR-140-F:5′-CGCAGTGGTTACCCTA-3′: miR-140-R: 5′-AGTGCAGG GTCCGAGGTATT-3′). The relative amount of each studied mRNA was normalized to GAPDH mRNA levels as housekeeping gene, and the data were analyzed according to the $2^{-\Delta\Delta CT}$ method.

## Statistical method

Data were analyzed using GraphPad Prism 8 Software and were presented as mean ± standard error (SE). F test or the Brown-Forsythe test was used for testing the equality of variances. Results were analyzed using unpaired Student's t-test for comparison between two groups. For multiple comparisons, results were analyzed using one-way or two-way ANOV A followed by Turkey post-test analysis. Results were considered significant at *p<0.05.

## Result

### Effects of DFMG on angiogenesis in the early stage of AS

A cellular model of oxidative stress-induced endothelial injury in HUVE-12 cells was successfully established and applied to mimic early changes in AS. HUVE-12 cells were treated with 5, 10, 20, 30, 50, 100μM LPC for 24 hours. The degree of LDH released into the supernatant was used to assess the degree of oxidative damage in HUVE-12 cells, as measured by the LDH kit. We found that LPC induced cell injury in a concentration dependent manner, but it affected cell proliferation in a different pattern: it promoted proliferation of HUVE-12 cells at concentrations less than 50μM (Fig 1A). Therefore, in following experiments, 30μM LPC was selected to induce damage to HUVE-12 cells.

Compared with the LPC group in which no intervention was performed, LDH release was reduced and cell proliferation activity was elevated in the LPC induced oxidative damage cell model when treated with 3.0μM DFMG (P < 0.05, Fig 1B). It suggested that 3.0μM DFMG can reduce the oxidative stress damage induced by 30μM LPC in HUVE-12 cells. Meanwhile, the results showed that the ability of tubeformation and CAM angiogenesis of HUVE-12 cells in LPC group was higher than in NC group, while that in DFMG group was lower than in LPC

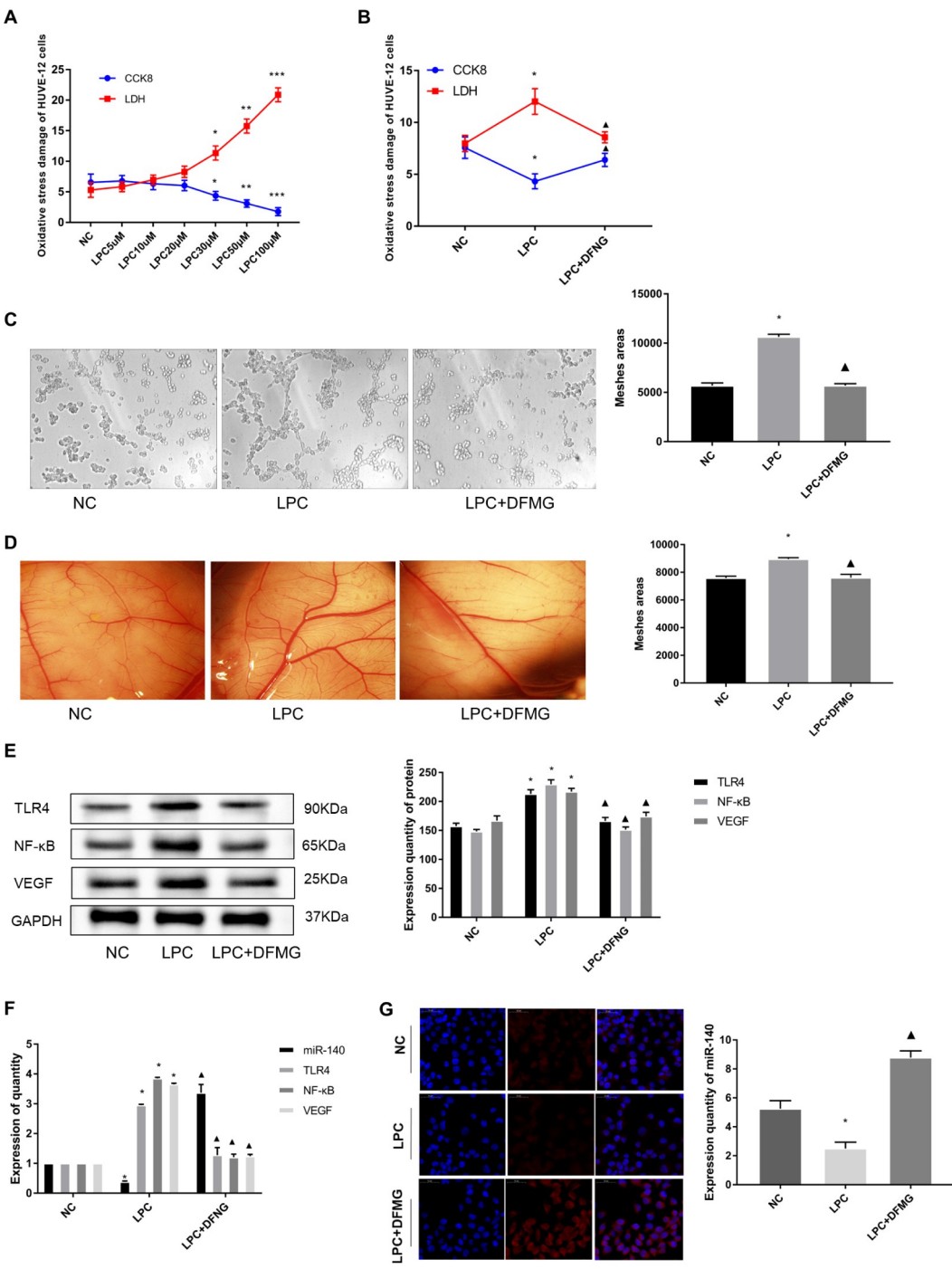

**Fig 1. DFMG attenuates angiogenesis induced by oxidative stress injury in endothelial cells.** HUVE-12 cells were treated with 30 μ M LPC for 24 hours, and the LPC+DFMG group was pre-incubated with 3.0 μ M DFMG for half an hour before adding LPC treatment, and then 30 μ M LPC was added for 24 hours. **A:** HUVE-12 cells were treated with 5, 10, 20, 30, 50, 100 μ M LPC for 24 hours, and the LDH release and cell proliferation activity of HUVE-12 cells were detected by LDH and CCK8 kits, respectively. Compared with NC group, *: P<0.05, **: P<0.01, ***: P < 0.001; **B:** DFMG could reduce the oxidative stress injury of HUVE-12 cells. LDH and CCK8 kits were used to detect the LDH release and cell proliferation of HUVE-12 cells. *: Compared with NC group, P<0.05, ▲: Compared with LPC group, P<0.05; **C:** Detection of the ability of cell tubule formation by Matrigel assay, *: Compared with NC group, P<0.05, ▲: Compared with LPC group, P<0.05; **D:** Observation of angiogenesis by CAM experiment, *: Compared with NC group, P<0.05, ▲: Compared with LPC group, P<0.05; **E:** Detect the protein expression of TLR4,NFκ-B and VEGF in HUVE-12 cell, *: Compared with NC group, P<0.05,

▲: Compared with LPC group, P<0.05; **F**: RT-qPCR to detect the expression of miR-140, TLR4,NFκ-B and VEGF in cells, *: Compared with NC group, P<0.05, ▲: Compared with LPC group, P<0.05; **G**: Detection of miR-140 expression in HUVE-12 cells by cellular immunofluorescence, *: Compared with NC group, P<0.05, ▲: Compared with LPC group, P<0.05.

group (P < 0.05, Fig 1C and 1D). It is suggested that DFMG can reduce injury and angiogenesis induced by oxidative stress.

## Effects of DFMG on miR-140 expression and TLR4/NF- κB/VEGF signal pathway of HUVE-12 cells in oxidative stress-induced cellular model

HUVE-12 cells in DFMG groups were pretreated with 3.0μM DFMG for 2h and then co-cultured with 30μM LPC like other groups. RT-qPCR and IF results showed that cellular miR-140 expression was significantly lower in the LPC group compared with NC (P < 0.05) (Fig 1F and 1G). In addition, the expression of TLR4 and its downstream signaling pathway members NF-κB and VEGF were detected by WB and RT-qPCR, and the results showed that the expression of TLR4, NF-κB and VEGF were significantly higher in the LPC group compared with the NC group. (P<0.05) (Fig 1E and 1F). It indicates that miR-140 expression is decreased and TLR4, NF-κB, VEGF expression is increased in oxidatively stressed HUVE-12 cells. In contrast, miR-140 expression was increased and TLR4, NF-κB, and VEGF expression were decreased in the DFMG group compared with the LPC group, which demonstrate that DFMG could increase the expression of miR-140 and decrease the expression of TLR4, NF-κB, and VEGF in oxidative stress-damaged HUVE-12 cells.

## DFMG inhibited the activation of TLR4/NF- κB/VEGF signal pathway and reduced oxidative stress-induced angiogenesis by up-regulating the expression of miR-140

To explore whether the effect of DFMG on the angiogenesis induced by the oxidative stress was associated with TLR4, we constructed HUVE-12 cell lines with either stable TLR4 over-expression, andTLR4 knockdown. The efficiency of lentiviral transfection in HUVE-12 cells was over 90% under fluorescent microscope (Fig 2A). It was verified that the expression of TLR4 was significantly higher in the cell lines overexpressing TLR4 (P <0.05) or decreased significantly in the cell lines silencing the expression of TLR4 (P<0.05) compared with the vehicle group using RT-qPCR or Western blot analysis. Compared with the vehicle, the expression of miR-140 was higher in HUVE-12 cell transfected by miR-140 mimics (P<0.05) and lower in HUVE-12 cells transfected by miR-140 inhibitor (P<0.05) (Fig 2B and 2C).

TLR4-knockdown HUVE-12 cells were incubated with 30 μM LPC, which was the same treatment method as establishing an oxidative stress injury cell model, and did not show some damages, such as: LDH release did not increase, the viability of HUVE-12 cells did not decrease, and some downstream molecules of the TLR4 cascade, such as NF-κB and VEGF, were significantly decreased (P<0.05) (Fig 2D). These changes became more significantly obvious after the addition of 3.0μM DFMG. In contrast, in the TLR4 over-expression group, the phenomena mentioned above are all reversed. Compared with the vehicle group without intervention, whether in the TLR4 over-expression group or in the TLR4knockdown group, the expression of miR-140 detected by RT-qPCR in HUVE-12 cell did not change significantly (P<0.05), but was significantly reduced in the DFMG group in which HUVE-12 cells was pretreated with 3.0μM DFMG (P<0.05) (Fig 2E).

At the same time, we observed that: compared with HUVE-12 cells in the control group, the number of tubules in the tubule formation test and newly generated blood vessels in the

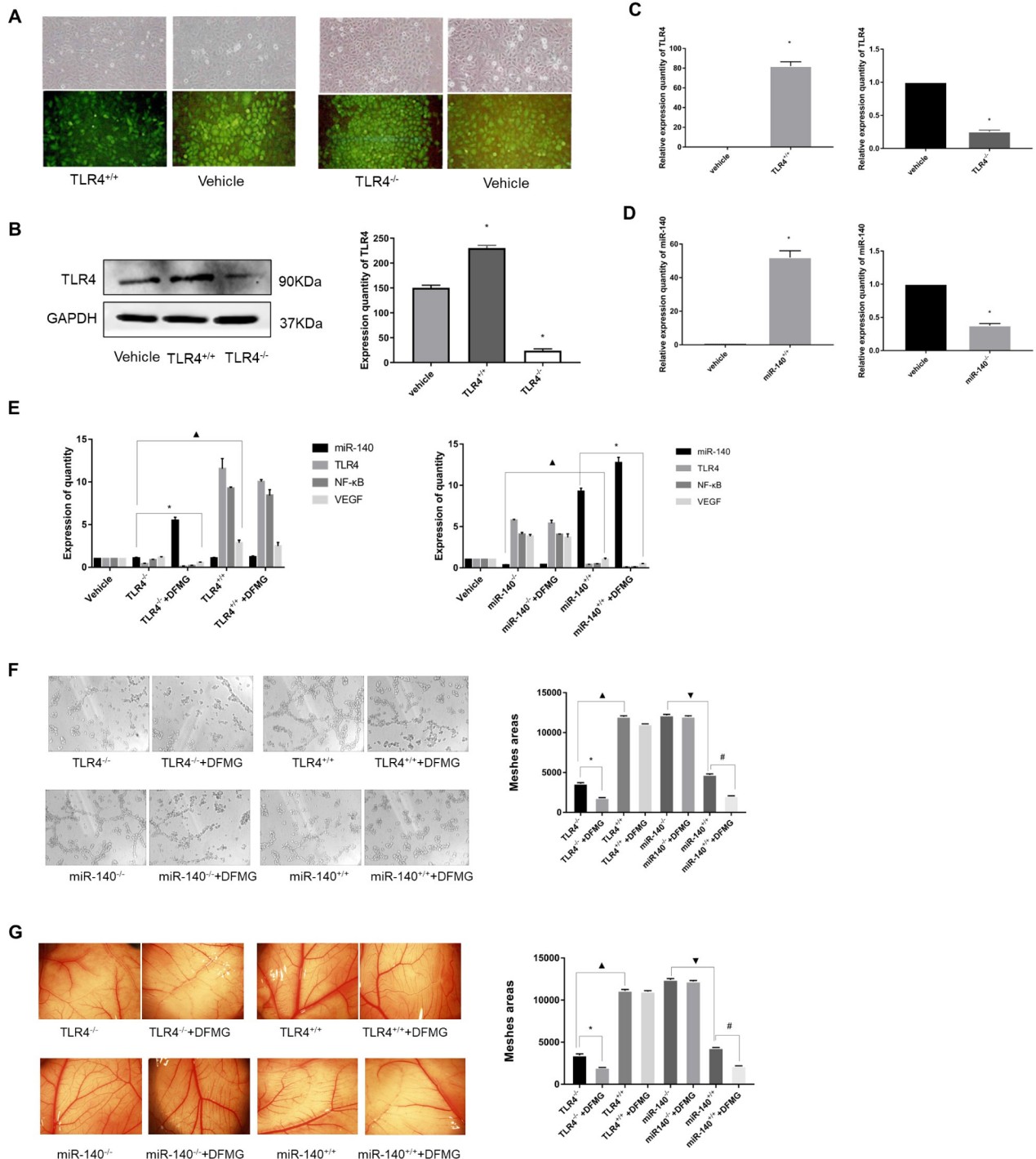

**Fig 2. DFMG inhibits the activation of TLR4/NF- κ B / VEGF signal pathway by up-regulating the expression of miR-140, and reduces angiogenesis of HUVE-12 cells under oxidative stress. A:** HUVE-12 cells were treated with recombinant lentivirus mixture (titer $1 \times 10^8$TU/mL, lentivirus 5uL) for 72 hours. The cells were cultured and screened in the complete medium supplemented with puromycin (2.5ug/mL). The transfection efficiency was observed under fluorescence microscope. The transfection efficiency of cells was observed under fluorescence microscope, and the number of cells with fluorescence was more than 90%. **B:** The expression of TLR4 protein in HUVE-12 cells was detected by WB, *: compared with the Vehicle group, P<0.05. **C:** The expression of TLR4 mRNA in HUVE-12 cells was detected by RT-qPCR, *: compared with the Vehicle group, P<0.05. **D:** Transient transfection of miR-140 mimics,inhibitor and negative control (NC) by lipo2000, after the cells were treated with plasmid mixture for 48 hours, the expression of miR-140 was detected by RT-qPCR., *: compared with the Vehicle group, P<0.05. **E:** RT-qPCR was used to observe the effects of DFMG on the expression of TLR4, NF- κ B, VEGF mRNA and miR-140 in HUVE-12 cells silently expressing or overexpressing TLR4 or miR-140, *: P<0.05. **F:** Detection of the ability of cell tubule formation by Matrigel assay, *: P<0.05. **G:** Observation of angiogenesis by CAM experiment, *: P<0.05.

chorioallantoic membrane were significantly decreased in the TLR4 knockdown HUVE-12 cells group (P<0.05) (Fig 2G). These changes were more significant in the HUVE-12 group with 3.0 μ M DFMG (P<0.05) (Fig 2G). In turn, in HUVE-12 group with overexpression of TLR4, the number of tubule formation and neovascularization were significantly increased which could not be reversed by 3.0 μ M DFMG. miR-140 can reduce the number of tubule formation and neovascularization induced by 30 μ M LPC. Compared with miR-140 mimics group, the number of tubule formation and newly blood vessels decreased significantly in mir-140-mimics + DFMG group (P<0.05) (Fig 2F). Compared with the LPC-induced model group, the number of tubule formation and neovascularization were increased in the miR-140 inhibition group which cannot be reversed by DFMG.

## Validation of the targeted regulatory relationship between miR-140 and TLR4

In this study, it was found that miR-140 may regulate the transcription of TLR4 by the bioin-formatics approaches such as Target Scan (http://www.targetscan.org). Further verification by dual-luciferase reporter assay showed that luciferase luminescence signal was decreased in the HUVE-pmirGLO-miR140-5p cells transfected with mutated TLR4, compared with in the HUVE-pmirGLO-miR140-5p cells transfected with mutated TLR4. This suggested hsa-miR-140-5p targeted directly with TLR4 (P<0.05) (Fig 3A).

Rescue experiments further verified the negative regulatory effect of miR-140 on TLR4. HUVE-12 cells stably over-expressed or knocked down TLR4 were transiently transfected by miR-140-5p inhibitor or miR-140-5p mimics by using Lipofectamine™ 2000 (lipo2000) for transfection, respectively. We found that the expression of TLR4 in the stably TLR4-overex-pression HUVE-12 cells increased after being transfected by miR-140-5p inhibitor, while it decreased significantly after transfection with miR-140-5p mimics (P<0.05). On the other hand, NF-κ B and VEGF, the downstream proteins of TLR4 signaling pathway showed a sig-nificant increase in TLR4-overexpressing HUVE-12 cells after transfection with miR-140-5p inhibitor (P<0.05), while their expression decreased upon transfection with miR-140-5p mim-ics (P<0.05). (P<0.05) (Fig 3B and 3C).

## Identification of ApoE$^{-/-}$ and TLR4$^{-/-}$/ ApoE$^{-/-}$ AS model mice

In order to investigate the function of TLR4 in the process of AS, we generated a TLR4$^{-/-}$/ApoE$^{-/-}$ mouse line in which both TLR4- and ApoE-coding region were knocked out. The DNA samples of mice were extracted and the genotypes of mice were identified by agarose gel electrophoresis. The results showed that 245bp was ApoE knockout mice, 155bp was wild type mice, 140bp was TLR4 and ApoE knockout mice, and heterozygous mice developed in both 140bp and 390bp (Fig 4A). This indicates that ApoE$^{-/-}$ and TLR4$^{-/-}$/ApoE$^{-/-}$ mice were success-fully constructed in this experiment. Different degrees of atherosclerotic plaque formation were found on the aortic wall of ApoE$^{-/-}$ and TLR4$^{-/-}$/ApoE$^{-/-}$ mice by somatograph, while no atherosclerotic plaque formation in the Wild Type (Fig 4B), indicating that the atherosclerotic animal model was successfully constructed.

TLR4 knockout combined with DFMG intervention significantly reduced blood lipids, pla-que formation and lipid infiltration in AS model ApoE$^{-/-}$ mice. Mice were fed with high fat and intervened with DFMG for 16 weeks. Results from automatic biochemical analyzer dem-onstrated a statistically significant increase in serum lipoproteins in ApoE$^{-/-}$ mice as compared with wild type C57 mice. Compared with ApoE$^{-/-}$ mice, serum lipoproteins in ApoE$^{-/-}$ mice treated with DFMG were significantly reduced, and serum lipoproteins in TLR4$^{-/-}$ApoE$^{-/-}$ mice were also significantly reduced. We also observed that changes in atherosclerotic plaque

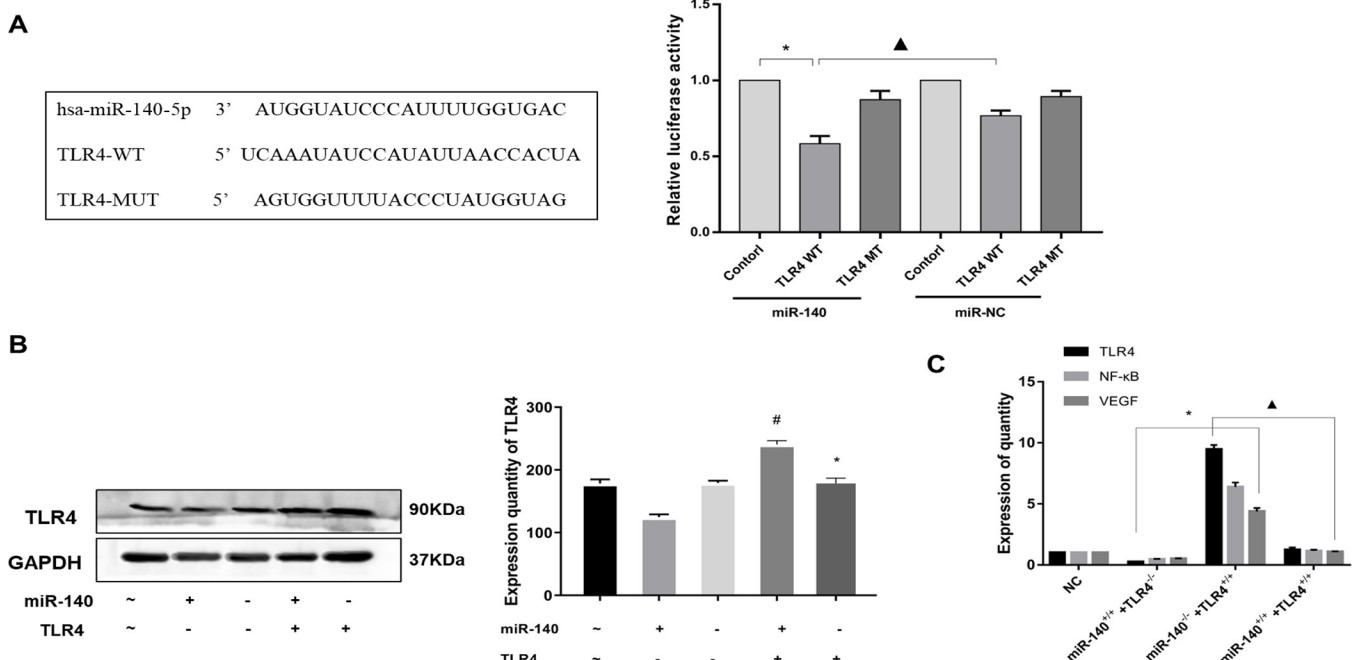

**Fig 3. Verification of the negative regulation relationship between miR-140 and TLR4 targeting. A:** hsa-miR-140-5p significantly down-regulated the reporter fluorescence of wild-type TLR4 vector. After mutating the predicted target site of TLR4, the down-regulation effect of hsa-miR-140-5p disappeared, while the report fluorescence of mutant TLR4 vector increased. *: Compared with miR-140-control group, P<0.05, ▲: Compared with miR-NC-TLR4 WT group, P<0.05. **B:** Detection of TLR4 protein expression by WB, *: Compared with miR-140[+/+]+TLR4[-/-] group, P<0.05, #: Compared with miR-140[-/-]+TLR4[+/+] group, P<0.05. **C:** The expression of TLR4, NF- κ B and VEGF mRNA was detected by RT-qPCR, *: Compared with miR-140[+/+]+TLR4[-/-] group, P<0.05, ▲: Compared with miR-140[-/-]+TLR4[+/+] group, P<0.05.

formation and lipid infiltration were substantially similar to changes in serum lipoproteins. (P<0.05) (Table 1, Fig 4B and 4C).

## DFMG reduces angiogenesis and maintain plaque stability in atherosclerotic plaques of ApoE[-/-] mice

Then, we detected the expression of VEGF, VEGFR2 and vWF to analyze the effect of DFMG on angiogenesis in atherosclerotic plaque using immunohistochemistry. The results showed that the expression of VEGF, VEGFR2 and vWF in ApoE[-/-] group was significantly higher than that in WT group, while the expression of VEGF, VEGFR2 and vWF showed a significant decrease after intervention with DFMG. Similarly, the expression of VEGF, VEGFR2 and vWF was reduced in the TLR4[-/-]ApoE[-/-] group compared with the ApoE[-/-] group (P < 0.05) (Fig 5E). We detected the collagen fiber content in the plaque to analyze the stability of plaque by Masson staining. The results showed the collagen fiber content in plaque of aortic arch in ApoE[-/-] group was significantly lower than that in TLR4[-/-]ApoE[-/-] group, while the collagen fiber content in 3.0μM DFMG + ApoE[-/-] group was significantly higher than that in ApoE[-/-] group (P < 0.05) (Fig 5F). It is suggested that DFMG could reduce angiogenesis and contribute to atherosclerotic plaque stability in ApoE[-/-] mice.

## DFMG increases the expression of miR-140 and inhibits the activation of TLR4/NF-κB/VEGF signal pathway in AS model mice

In this study, the negative targeting regulation relationship between TLR4 and miR-140 has been confirmed in vitro. In order to further explore this regulatory mechanism, we detected

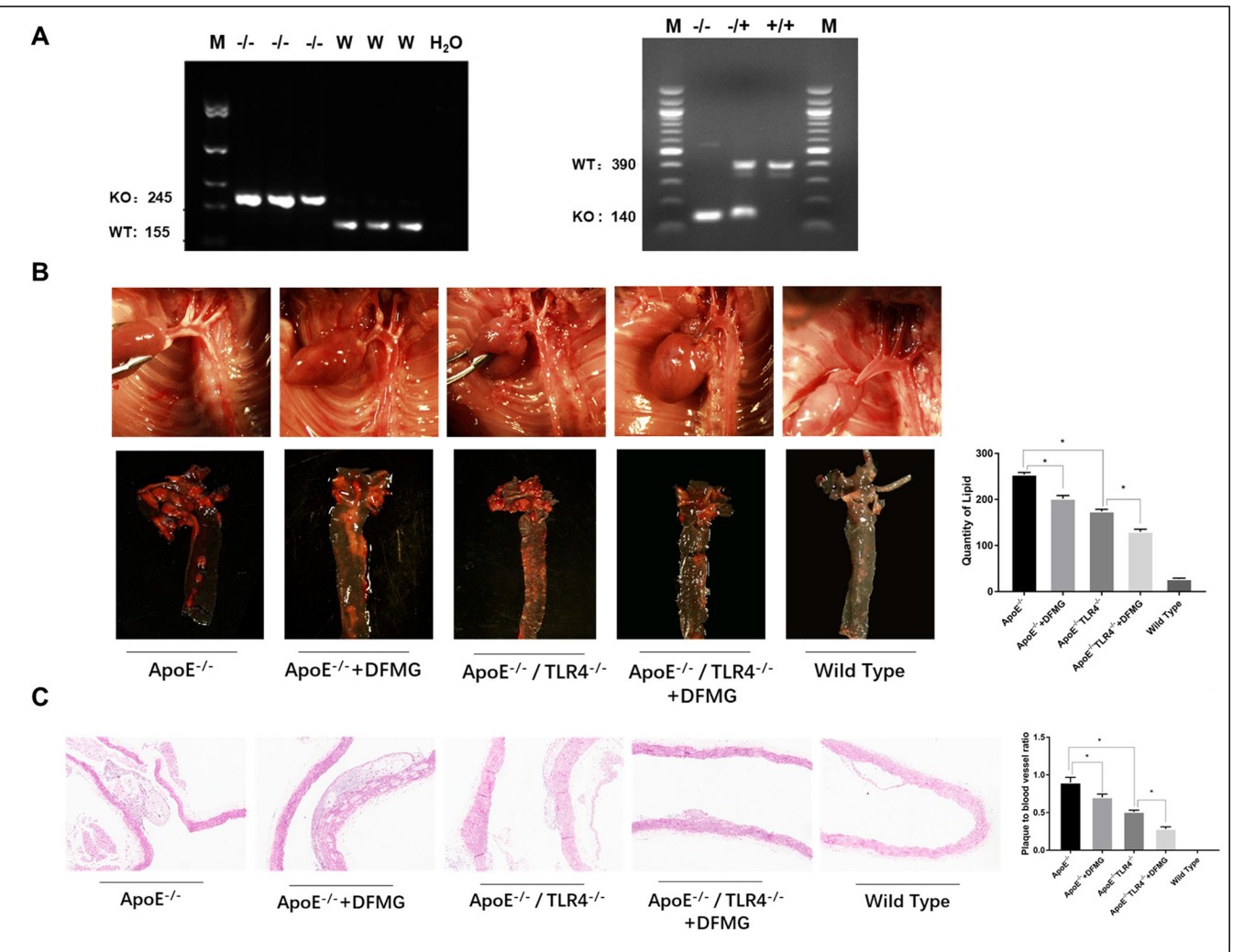

**Fig 4. DFMG reduces AS plaque formation and lipid infiltration in AS model mice.** ApoE-/- mice and ApoE$^{-/-}$/TLR4$^{-/-}$ mice were fed with high-fat diet for 16 weeks. DFMG group mice were fed with high fat and supplemented with DFMG (10mg/ (kg ·d)). C57BL/6 mice were used as blank control group (Wild Type group). Wild Type group was fed with routine diet without any drug intervention, with 10 mice in each group. **A:** Detection of DNA in mice by agarose gel electrophoresis, the results showed that 245bp was ApoE knockout mice, 155bp was wild type mice, 140bp was TLR4 and ApoE knockout mice, and heterozygous mice developed in both 140bp and 390bp. **B:** The plaque formation of thoracic aorta in mice was observed under gross microscope and the lipid infiltration in the plaque was detected by oil red O staining. *: P<0.05. **C:** HE staining was used to observe the histopathology of AS plaque in mouse thoracic aorta, and the ratio of plaque to blood vessel was measured and calculated. *: P<0.05.

the expression of TLR4, NF- κB, VEGF and miR-140 in each group of mice. The results showed that miR-140 expression significantly decreased and TLR4, NF-κB and VEGF expression increased in the ApoE$^{-/-}$ mice compared with the WT group (P< 0.05). miR-140 expression increased and TLR4, NF-κB, and VEGF expression decreased in the 3.0μM DFMG-treated ApoE$^{-/-}$ group compared with the ApoE$^{-/-}$ group (P<0.05). Moreover, miR-140 expression increased and TLR4, NF-κB and VEGF expression decreased in the TLR4$^{-/-}$ApoE$^{-/-}$ group compared to the ApoE$^{-/-}$ group (P<0.05) (Fig 5A, 5B and 5D), furthermore, it was found that there was a negative correlation between miR-140 and TLR4 (r = 0.8151, P< 0.05) (Fig 5C).

This study revealed that DFMG) inhibited angiogenesis induced by oxidatively damaged endothelium. Studies conducted on high-fat fed ApoE$^{-/-}$ mice showed that DFMG lowered the

**Table 1. Blood lipid levels and body weight of AS mice.**

| Group | HDL | LDL | TG | CHOL | VLDL | Body Weight(g) |
|---|---|---|---|---|---|---|
| | (mmol/L) | (mmol/L) | (mmol/L) | (mmol/L) | (g) | |
| Wild type | 2.11±0.31 | 0.78±0.13 | 0.27±0.07 | 2.86±0.81 | 26.18±0.89 | 24.33±1.12 |
| ApoE$^{-/-}$ | 2.10±0.22* | 6.15±0.41* | 1.78±0.13* | 19.54±1.24* | 28.84±0.49* | 27.28±1.23* |
| ApoE$^{-/-}$+DFMG | 2.46±0.11# | 4.77±0.22# | 1.21±0.17# | 16.77±1.35# | 25.43±0.69# | 26.16±1.21# |
| ApoE$^{-/-}$/TLR4$^{-/-}$ | 2.31±0.21# | 4.05±1.61# | 1.31±0.14# | 14.56±0.55# | 25.17±0.57# | 25.88±1.32# |
| ApoE$^{-/-}$/TLR4$^{-/-}$+DFMG | 2.58±0.33▲ | 3.51±1.12▲ | 0.52±0.06▲ | 5.89±1.02▲ | 23.76±0.71▲ | 24.56±1.46▲ |

(* indicates comparison with Wild Type group, $P<0.05$

# indicates comparison with ApoE$^{-/-}$ group, $P<0.05$, ▲ indicates comparison with ApoE$^{-/-}$ TLR4$^{-/-}$group, $P<0.05$)

blood lipids and attenuated angiogenesis formation within the atherosclerotic plaques, thereby maintaining plaque stability.

## Discussion

This study reveals that DFMG possesses the capacity to mitigate endothelial oxidative stress damage, prevent the formation of atherosclerotic plaques, inhibit lipid infiltration and angiogenesis within these plaques, and maintain plaque stability. Its mechanism of action against atherosclerosis is closely associated with the targeted negative regulation of TLR4 through increased miR-140 expression, and the suppression of TLR4/NF-κB/VEGF signaling pathway activation. Oxidative stress-induced endothelial cell damage has been considered a key factor in the onset of AS and a crucial factor that directly promotes intraplaque angiogenesis [22]. This occurs during the formation of the vulnerable plaques and accelerates plaque instability, which can induce acute cardiovascular events like thromboembolism, stroke, etc [10, 23]. In this study, we successfully established a model of oxidative stress-induced endothelial cell damage by treating HUVE-12 cells with 30 μM LPC. Oxidative stress enhanced endothelial cell tubule formation and significantly increased the number of angiogenesis in the chicken chorioallantoic membranes. Therefore, this model could successfully simulate the angiogenesis response triggered by oxidative damage to the endothelial cells within AS vessels.

*In vitro* Matrigel and CAM assays were performed to study angiogenesis. In the Matrigel assay, oxidative stress-damaged HUVE-12 cells showed a significant increase in tubule forming capacity. The supernatant of oxidative stress-damaged HUVE-12 cells could significantly promote angiogenesis on CAM membranes. For *in vivo* studies, ApoE$^{-/-}$ mice were fed a high-fat diet for 16 weeks. Stereomicroscope observations and hematoxylin and eosin staining highlighted a varying degree of AS plaques in the aortic wall of AS mice model. To further observe intraplaque angiogenesis, the expression levels of relevant protein markers such as VEGF, VEGFR2, and vWF were measured using immunohistochemistry and immunofluorescence. The results revealed a significant increase in VEGF, VEGFR2, and vWF expression in the plaques of the ApoE$^{-/-}$ AS mice model. Further, the Masson staining results indicated a significant decrease in the content of intraplaque collagen fibers. Together, these results confirmed that the AS mice model was successfully established. In addition, increased intraplaque angiogenesis of the thoracic aortic plaques of the mice led to decreased plaque stability. Our results revealed that DFMG, a new chemical entity derived from the lead compound genistein, could attenuate oxidative stress-induced endothelial cell damage, reduce AS plaque formation and lipid infiltration, decrease intraplaque angiogenesis, and maintain plaque stability [18, 19]. Endothelial cells treated with 3.0 μM DFMG attenuated cellular oxidative stress-induced damage and tubule formation and were pro-angiogenesis. These results indicate that DFMG could

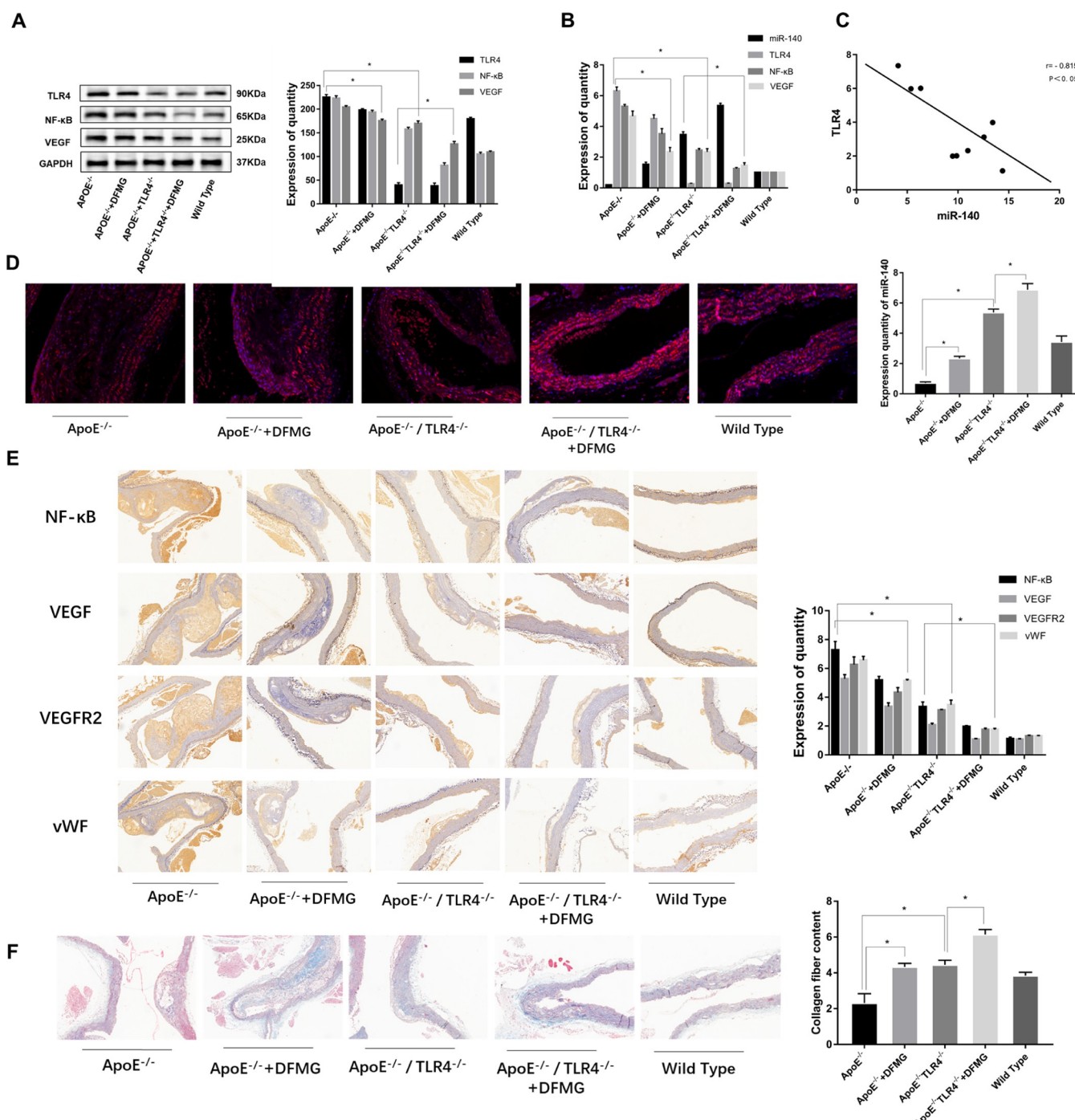

**Fig 5. DFMG suppresses the activation of the TLR4/NF-κB/VEGF signaling pathway by upregulating the expression of miR-140 in atherosclerosis (AS) model mice.** This contributes to a reduction in angiogenesis and the preservation of plaque stability in the AS plaques of model mice. **A:** Effect of DFMG on TLR4, NF-κB and VEGF protein expression in AS mice detected by WB, *: P<0.05. **B:** Effect of DFMG on miR-140, TLR4, NF-κB and VEGF expression in AS mice detected by RT-qPCR, *: P<0.05. **C:** There was negative correlation between miR-140 and TLR4 in AS mice, r = -0.8151, P<0.05. **D:** The effect of DFMG on the expression of miR-140 in mice was detected by IF, with blue as DAPI and red as the target staining. *: P<0.05. **E:** The effects of DFMG on the expression of NF-κ-B, VEGF, VEGFR2 and vWF in mice were detected by IHC, with hematoxylin staining in blue and DAB staining in brown. *: P<0.05. **F:** Masson staining was used to observe the effect of DFMG on the content of collagen fibers in mouse aortic atherosclerotic plaques, with collagen fibers in blue and muscle fibers in red, *: P<0.05.

attenuate oxidative stress-induced endothelial cell damage and consequent angiogenesis. *In vivo* studies have shown abnormal lipid levels caused by hyperlipidemia resulted in the deposition of large amounts of lipid under the arterial endothelium matrix. This triggered AS formation mediated by endothelial cells and chemokines. Intraplaque lipid infiltration aggravated the inflammatory responses and hypoxia, thereby promoting oxidative stress-induced endothelial cell damage, which releases abundant reactive oxygen species and upregulated pro-angiogenic factors, thereby inducing angiogenesis [24, 25]. Therefore, the lipid levels in mice were examined. The results revealed a significant increase in lipid levels in the ApoE$^{-/-}$AS mice model. Further, the lipid levels were reduced on treatment with DFMG. Additionally, DFMG reduced atherosclerotic plaque formation and lipid infiltration in the thoracic aorta of the ApoE$^{-/-}$ mice model, decreased intraplaque angiogenesis, and enhanced plaque stability.

Our group previously reported that the DFMG mediates the anti-AS effect by regulating Toll-like receptor 4 (TLR4), a member of the natural pattern recognition receptor family [26, 27]. TLR4 is a predominant receptor that mediates endotoxin lipopolysaccharide-induced immune responses and activates the NF-κB pathway by recruiting myeloid differentiation factor 88 (MYD88) [28, 29]. It is also associated with AS progression using various mechanisms [30]. microRNAs (miRNA) are a class of evolutionarily conserved non-coding single-stranded small RNA molecules which post-transcriptionally degrades target gene mRNAs or inhibit gene translation [31, 32]. miR-140 can regulate TLR4, which stimulates the body to produce inflammatory mediators via MyD88 signaling, thereby inducing innate immune responses [33, 34]. Therefore, we speculated that DFMG regulated AS angiogenesis via miR-140, which targeted TLR4. The results showed an increase in TLR4 expression in oxidative stress-damaged HUVE-12 cells induced by LPC. Further, an increase in the expression of TLR4 downstream signaling pathway, including NF-κB and VEGF, was observed, whereas a decrease in miR-140 expression was observed. After DFMG treatment, a significant reduction in TLR4, NF-κB, and VEGF expression was observed in oxidative stress-damaged HUVE-12 cells, whereas the miR-140 expression increased. DFMG had a weaker effect on TLR4 over-expression group compared to the TLR4-knockdown group in terms of oxidative damage and tubule formation in blood vessel endothelium. Further, the relationship between miR-140 and TLR4 was determined using a dual-luciferase reporter assay. HUVE-12 cells overexpressing TLR4 had enhanced oxidative stress-induced damage, increased endothelial cell tubule formation, and pro-angiogenic capacity. Silencing TLR4 expression alleviated oxidative stress-induced damage to endothelial cells and decreased endothelial cell tubule formation and angiogenic capacity. Upregulating miR-140 expression in TLR4 overexpressing HUVE-12 cells significantly reduced TLR4 expression. These results indicated that miR-140 could attenuate oxidative stress-induced damage and reduce cellular tubule formation and angiogenesis by negatively regulating TLR4 and inhibiting the TLR4/ NF-κB /VEGF signaling pathway. Similarly, TLR4 knockout ApoE$^{-/-}$AS mice showed increased miR-140 expression and decreased activation of the TLR4/ NF-κB /VEGF signaling pathway. This suggests a negative correlation between miR-140 and TLR4 expression. In HUVE-12 cells, the effect of DFMG on TLR4 expression was lost by silencing miR-140. Further, an increase in expression of downstream signaling pathways like NF-κB and VEGF was observed, along with enhanced endothelial cell tubule formation and angiogenesis. Interestingly, on treatment with miR-140-mimics, a significant increase in the inhibitory effect of DFMG on cellular TLR4, NF-κB, and VEGF expression was observed, along with a reduction in endothelial cell tubule formation and angiogenesis.

## Conclusions

In summary, DFMG demonstrates the capacity to alleviate oxidative stress-induced endothelial cell damage, hinder the formation of atherosclerotic plaques, impede intraplaque lipid

infiltration and angiogenesis, and uphold plaque stability. The anti-atherosclerotic effects of DFMG are mediated through the negative regulation of TLR4 by miR140, ultimately inhibiting the TLR4/NF-κB/VEGF signaling pathway. Furthermore, DFMG treatment shows potential in reducing the incidence of acute cardiovascular events, holding clinical significance. Nevertheless, further studies are essential to delve into its mechanisms of action and functionalities.

## Supporting information

**S1 Fig. Flowchart of animal experiment.** iG: intragastric infusion.
(TIF)

**S2 Fig. Structure diagram of plasmid vector.** (A:GV268, B:GV249, C: GV358, D: Plko.1 puro-GFP).
(TIF)

**S1 Raw images.**
(PDF)

**S1 Data.**
(XLSX)

## Acknowledgments

We thank the Institute of Model Animal, Wuhan University (IMA).

## Author Contributions

**Data curation:** Pingjuan Bai, Xueping Xiang, Jiawen Kang.

**Funding acquisition:** Xiaohua Fu, Yong Zhang.

**Methodology:** Pingjuan Bai, Xiaoqing Xiang, Jingwen Jiang.

**Writing – original draft:** Pingjuan Bai.

**Writing – review & editing:** Xiaohua Fu, Yong Zhang, Lesai Li.

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
