## [Decision Letter · Decision Letter 0]

14 Nov 2023

PONE-D-23-29456DFMG reduces the angiogenesis to maintain the plaque stability by inhibiting TLR4/VEGF pathway in atherosclerosisPLOS ONE

Dear Dr. Bai,

Thank you for submitting your manuscript to PLOS ONE. After careful consideration, we feel that it has merit but does not fully meet PLOS ONE’s publication criteria as it currently stands. Therefore, we invite you to submit a revised version of the manuscript that addresses the points raised during the review process.

We look forward to receiving your revised manuscript.

Kind regards,

Arvin Haj-Mirzaian

Academic Editor

PLOS ONE

Journal Requirements:

 " This study is supported by the Natural Science Foundation of China (grant. no. 81370382), and Natural Science Foundation of Hunan Province (grant. no. 14JJ2059 and 2022JJ30415). ZY and LLS are authors who received awards."

6. We notice that your supplementary figures are uploaded with the file type 'Figure'. Please amend the file type to 'Supporting Information'. Please ensure that each Supporting Information file has a legend listed in the manuscript after the references list.

**Additional Editor Comments:**

Please add ethical section and prepare all below information:

Ethical Considerations:

All research must adhere to the highest ethical standards. For studies involving animal models, it is imperative that all efforts be made to minimize suffering and reduce the number of animals used, and to justify the species and number of animals used. Authors must clearly state that they have conducted their studies ethically, and they should be prepared to provide evidence of ethical approval by an appropriate committee.

Animal Studies:

For this paper authors must declare that the study was conducted in compliance with the Animal Research: Reporting of In Vivo Experiments (ARRIVE) guidelines. The ARRIVE guidelines provide a checklist of information that should be included in publications reporting animal research, with the aim of improving the reporting standards and ensuring that the data can be thoroughly evaluated.

The ARRIVE guidelines encompass the following key areas:

Study design

Sample size

Animal characteristics

Housing and husbandry

Experimental procedures

Outcome measures

Statistical analysis

Euthanasia Method:

In accordance with the ARRIVE guidelines and ethical treatment of animals, authors must specify the method used for euthanasia of mice. This should include details such as the type of agent used (e.g., barbiturate overdose, carbon dioxide inhalation), the concentration and volume of the agent (where applicable), the method of administration, and any measures taken to ensure that euthanasia is performed as humanely as possible. Justification should be provided for the chosen method, referencing the most recent guidelines for humane killing provided by organizations such as the American Veterinary Medical Association (AVMA) or equivalent bodies.

Authors should also ensure that the personnel performing euthanasia are adequately trained and competent in the procedures used.

Ethical Approval:

Authors must state explicitly that the research protocol was approved by an institutional animal care and use committee (IACUC) or equivalent body. The name of the ethics committee that approved the study and the reference number/protocol ID should be provided.

Conflict of Interest:

Authors must declare any potential conflicts of interest relating to their research.

Reviewers' comments:

Reviewer's Responses to Questions

**Comments to the Author**

1. Is the manuscript technically sound, and do the data support the conclusions?

Reviewer #1: Yes

2. Has the statistical analysis been performed appropriately and rigorously? 

Reviewer #1: Yes

3. Have the authors made all data underlying the findings in their manuscript fully available?

Reviewer #1: Yes

4. Is the manuscript presented in an intelligible fashion and written in standard English?

Reviewer #1: Yes

5. Review Comments to the Author

Reviewer #1: 1-The title could be changed to "DFMG possibly reduces the angiogenesis …" with this small sample size.

2-Please explain " ImageJ" to the readers who are not familiar with it.

3-In many cases, excessive use of abbreviations can confuse and alienate unfamiliar audiences, and even well-intentioned writers and speakers may overestimate an audience's familiarity with abbreviations. Although abbreviations should not be avoided entirely, using them as a default can be problematic. This manuscript is full of familiar and unfamiliar abbreviations. You should explain each of your abbreviations the first time it appears in the main text, but have not any explanation about some abbreviations, such as miR-140, CCK8, LDH, RPMI-1640, WB, RT-qPCR, lipo2000, IHC, WT group, …. It is one of the rules of providing information for your readers.

4-In contrast, after the first appearance of the abbreviation, the abbreviation should always be used in the rest of the manuscript instead of the complete term. However, you re-explained "DFMG" in the last paragraph of 3.7 or "AS" in the discussion section, line 2.

5-In general, an abbreviation should be used if the term appears at least five times in the main text (The Abstract does not count). If you use the term or phrase only two, three or four times, it should not be abbreviated.

6-What is the meaning of "blank group"?

7-Where is "Table 1"?

8-The methodology needs to be clarified. You explained "methodology" in the "discussion" section; unfamiliar readers (such as clinicians) are confused about in vivo or in vitro study, the number of groups, the sample size in each group, the rational of any particular type of laboratory study, …, before reading discussion. For instance, you should mention the purpose of each study after you did it for your readers because readers may have a different line of attitude from you.

6. PLOS authors have the option to publish the peer review history of their article (what does this mean?). If published, this will include your full peer review and any attached files.

Reviewer #1: No

---

## [Author Response · Author response to Decision Letter 0]

1 Jan 2024

Response to Reviewers:

Dear Reviewers,

Thank you very much for your time involved in reviewing the manuscript and your very encouraging comments on the merits. 

We also appreciate your clear and detailed feedback and hope that the explanation has fully addressed all of your concerns. In the remainder of this letter, we discuss each of your comments individually along with our corresponding responses.

To facilitate this discussion, we first retype your comments in italic font and then present our responses to the comments.

1-The title could be changed to "DFMG possibly reduces the angiogenesis …" with this small sample size.

Response 1: Thank you very much for your suggestion, the title of the article has been modified.

2-Please explain " ImageJ" to the readers who are not familiar with it.

Response 2: We add an explanation for “ImageJ”, The relevant contents are provided below as a screen dump for your quick reference.

3-In many cases, excessive use of abbreviations can confuse and alienate unfamiliar audiences, and even well-intentioned writers and speakers may overestimate an audience's familiarity with abbreviations. Although abbreviations should not be avoided entirely, using them as a default can be problematic. This manuscript is full of familiar and unfamiliar abbreviations. You should explain each of your abbreviations the first time it appears in the main text, but have not any explanation about some abbreviations, such as miR-140, CCK8, LDH, RPMI-1640, WB, RT-qPCR, lipo2000, IHC, WT group, …. It is one of the rules of providing information for your readers.

Response 3: Thank you for the detailed review. We have carefully and thoroughly proofread the manuscript to correct all the abbreviations.

4-In contrast, after the first appearance of the abbreviation, the abbreviation should always be used in the rest of the manuscript instead of the complete term. However, you re-explained "DFMG" in the last paragraph of 3.7 or "AS" in the discussion section, line 2.

Response 4: Thank you for the detailed review. We have carefully and thoroughly proofread the manuscript to correct all the abbreviations.

5-In general, an abbreviation should be used if the term appears at least five times in the main text (The Abstract does not count). If you use the term or phrase only two, three or four times, it should not be abbreviated.

Response 5: Thank you very much for your advice, which has been modified in the article.

6-What is the meaning of "blank group"?

Response 6: This is a translation error, which is modified in the article as “Vehicle group” and “Wild Type”, respectively.

7-Where is "Table 1"?

Response 7: We uploaded Table 1 in the article

8-The methodology needs to be clarified. You explained "methodology" in the "discussion" section; unfamiliar readers (such as clinicians) are confused about in vivo or in vitro study, the number of groups, the sample size in each group, the rational of any particular type of laboratory study, …, before reading discussion. For instance, you should mention the purpose of each study after you did it for your readers because readers may have a different line of attitude from you.

Response 8: Thanks for your great suggestion on improving the accessibility of our manuscript. We add the title of the research purpose in each method section.

We would like to take this opportunity to thank you for all your time involved and this great opportunity for us to improve the manuscript. We hope you will find this revised version satisfactory.

Sincerely,

The Authors: Pingjuan Bai

Response to Editor:

Dear Arvin Haj-Mirzaian,

Thank you for your prompt and constructive feedback on my manuscript entitled " DFMG may potentially decrease angiogenesis to preserve plaque stability by inhibiting the TLR4/VEGF pathway in atherosclerosis," which I submitted to PLOS ONE. I appreciate the time and effort you and the reviewers have invested in evaluating my work.

I have carefully considered each of your comments and made the necessary revisions to address the concerns raised. Below, I outline the changes made in response to your suggestions:

Response 1: I have made revisions according to the formatting requirements of the PLOS ONE journal.

Response 2: Thank you for your valuable advice. However, I have not yet uploaded the raw experimental data to the repository at the moment.

 " This study is supported by the Natural Science Foundation of China (grant. no. 81370382), and Natural Science Foundation of Hunan Province (grant. no. 14JJ2059 and 2022JJ30415). ZY and LLS are authors who received awards."

Response 3: This study is supported by the Natural Science Foundation of China (Grant No. 81370382) and the Natural Science Foundation of Hunan Province (Grant No. 14JJ2059 and 2022JJ30415). Xiaohua Fu and Yong Zhang, recipients of awards. Their contribution to this article involved reviewing and editing.

Response 4: We have incorporated an ethical statement and supplied the complete name and code of the Ethics Committee for the "mice" section in the Methods.

Response 5: The blot/gel images in this article have been uploaded in the required PDF format as requested. 

6. We notice that your supplementary figures are uploaded with the file type 'Figure'. Please amend the file type to 'Supporting Information'. Please ensure that each Supporting Information file has a legend listed in the manuscript after the references list

Response 6: We have made revisions and re-uploaded.

The following are comments in response to other editors:

All procedures involving animals were conducted in strict accordance with ARRIVE guidelines for the care and use of laboratory animals and were approved by the Ethics Committee of Hunan Normal University (Approval No. 2013289). Researchers engaged in animal experimentation possess the Laboratory Animal Professional Post qualification, verified by the certificate code 20180054.

The -7-week-old SPF-grade ApoE gene knockout mice and the 6-7-week-old SPF-grade TLR4 gene knockout mice were graciously donated by the Collaborative Innovation Center for Model Animal at Wuhan University (Animal Qualification Testing Report Number: BJYRL-WBKH-20160224A1). The ApoE and TLR4 double gene knockout mice were bred by our research team through the mating of the aforementioned gene knockout mice. Non-transgenic mice with the same genetic background, aged 5-6 weeks (SPF-grade C57 BL/6 mice), were procured from Hunan Slake Company. All experimental mice were housed in the SPF-grade animal facility at the School of Medicine, Hunan Normal University, maintaining a 12-hour light/12-hour dark cycle. More detailed experimental methods and schematic diagrams can be found in the Methods section of the manuscript and in the S1 figures.

Conflict of Interest: The authors declare that they have no known competing financial interests or personal relationships that could have appeared to influence the work reported in this paper.

I believe these revisions have strengthened the manuscript, improving its overall clarity and coherence. I trust that these changes align with the expectations of PLOS ONE and enhance the quality of the manuscript.

Please find the revised manuscript attached. I hope these modifications meet the standards of the journal. If you require any further clarification or additional information, please do not hesitate to contact me.

Thank you once again for your valuable guidance throughout this process. I look forward to hearing from you soon.

Sincerely,

The Authors: Pingjuan Bai

---

## [Decision Letter · Decision Letter 1]

12 Feb 2024

PONE-D-23-29456R1DFMG may potentially decrease angiogenesis to preserve plaque stability by inhibiting the TLR4/VEGF pathway in atherosclerosisPLOS ONE

Dear Dr. Bai,

Thank you for submitting your manuscript to PLOS ONE. After careful consideration, we feel that it has merit but does not fully meet PLOS ONE’s publication criteria as it currently stands. Therefore, we invite you to submit a revised version of the manuscript that addresses the points raised during the review process.

We look forward to receiving your revised manuscript.

Kind regards,

Arvin Haj-Mirzaian

Academic Editor

PLOS ONE

Journal Requirements:

Reviewers' comments:

Reviewer's Responses to Questions

**Comments to the Author**

1. If the authors have adequately addressed your comments raised in a previous round of review and you feel that this manuscript is now acceptable for publication, you may indicate that here to bypass the “Comments to the Author” section, enter your conflict of interest statement in the “Confidential to Editor” section, and submit your "Accept" recommendation.

Reviewer #1: All comments have been addressed

Reviewer #2: All comments have been addressed

2. Is the manuscript technically sound, and do the data support the conclusions?

Reviewer #1: Yes

Reviewer #2: Yes

3. Has the statistical analysis been performed appropriately and rigorously? 

Reviewer #1: Yes

Reviewer #2: Yes

4. Have the authors made all data underlying the findings in their manuscript fully available?

Reviewer #1: Yes

Reviewer #2: Yes

5. Is the manuscript presented in an intelligible fashion and written in standard English?

Reviewer #1: Yes

Reviewer #2: Yes

6. Review Comments to the Author

Reviewer #1: 1- "… counted by ImageJ (ImageJ is a java-based … of Health (NIH)." can be revised as "… counted by ImageJ [ImageJ is a Java-based … of Health (NIH)]."

2- You mentioned, "We uploaded Table 1 in the article," but the location of "Table 1" is not specified.

3- Kindly provide an explanation regarding the number of groups and the corresponding sample size in each group.

Reviewer #2: Dear Editor,

On the manuscript entitled: DFMG may potentially decrease angiogenesis to preserve plaque stability by inhibiting the TLR4/VEGF pathway in atherosclerosis.

Comments:

This is an interesting study. Authors addressed comments raised by reviewers however some issues should be clarified before acceptance.

1- There are various typo errors in the text.

2- Mice add to the title

3- “Detection of blood lipids in mice” change to “Detection of blood lipids”

4- Real-time fluorescence quantitative PCR change to “quantitative Real-time PCR”

5- Method for anesthesia should be clarified. What did you mean from “The next day, the mice were anesthetized by abdominal cavity”?

6- Discussion should be began with your core finding.

7. PLOS authors have the option to publish the peer review history of their article (what does this mean?). If published, this will include your full peer review and any attached files.

Reviewer #1: No

Reviewer #2: No

---

## [Author Response · Author response to Decision Letter 1]

5 Mar 2024

respond to reviewer 

Dear Reviewer,

Thank you very much for your valuable comments and suggestions on our manuscript. We have carefully considered and revised the manuscript according to your guidance.

Regarding your points about the lack of specific location for figure files, inadequate explanation of group numbers, and inaccuracies in terminology and expressions, we have made revisions to the manuscript and provided explanations in the response letter. Additionally, we have corrected typos and grammatical errors in the text.

To facilitate this discussion, we first retype your comments in italic font and then present our responses to the comments.

Reviewer #1: 

1- "… counted by ImageJ (ImageJ is a java-based … of Health (NIH)." can be revised as "… counted by ImageJ [ImageJ is a Java-based … of Health (NIH)]."

Response：Thank you for such careful review, we have made modifications to the paper.

2- You mentioned, "We uploaded Table 1 in the article," but the location of "Table 1" is not specified.

Response：We apologize for the inconvenience. Table 1 was inadvertently uploaded to the submission system, and we have now included Table 1 in the paper.

3- Kindly provide an explanation regarding the number of groups and the corresponding sample size in each group.

Response：The groupings for the in vitro experiments have been incorporated into the methods section of the article, whereas detailed explanations of the groupings for the in vivo animal experiments are provided in Figure S1.

Reviewer #2

1- There are various typo errors in the text.

Response: Thank you for your meticulous review of the article, and we have rectified these errors accordingly.

2- Mice add to the title

Response： We changed the paper title to “DFMG decreases angiogenesis to uphold plaque stability by inhibiting the TLR4/VEGF pathway in mice.”

3- “Detection of blood lipids in mice” change to “Detection of blood lipids”

Response： Thank you for your suggestion, it has been changed to “Detection of blood lipids”.

4- Real-time fluorescence quantitative PCR change to “quantitative Real-time PCR”

Response： Thank you for your suggestion, it has been changed to “quantitative Real-time PCR”.

5- Method for anesthesia should be clarified. What did you mean from “The next day, the mice were anesthetized by abdominal cavity”?

Response： Thank you very much for your suggestion. The description has been revised in the article accordingly.

6- Discussion should be began with your core finding.

Response： Thank you for your valuable suggestions. We have incorporated a section in the discussion of the article highlighting the core findings of our study.

We appreciate your thorough review and helpful feedback. We believe that the revised manuscript is more refined, and we look forward to your further assessment.

Thank you once again for your time and expertise.

Best regards,

Sincerely,

The Authors: Pingjuan Bai

Respond to editor： 

Dear Arvin Haj-Mirzaian,

We would like to express our sincere gratitude to you and the reviewers for the constructive feedback and valuable suggestions provided for our manuscript entitled " DFMG decreases angiogenesis to uphold plaque stability by inhibiting the TLR4/VEGF pathway in mice"[ PONE-D-23-29456R1]. We have carefully considered all the comments and suggestions provided by the reviewers and have made the necessary revisions to improve the quality and clarity of the manuscript.

Journal Requirements:

Response：

After a thorough examination of the references section of our paper, we confirm that no citations of retracted papers were found. Furthermore, the format of the references adheres to the requirements of your esteemed journal.

We believe that these revisions have strengthened the manuscript and addressed the concerns raised by the reviewers. We hope that these changes meet the expectations of the journal and contribute to the improvement of the manuscript.

Thank you once again for the opportunity to revise and resubmit our manuscript. We look forward to hearing from you regarding the further processing of our submission.

Sincerely,

Pingjuan Bai

---

## [Editor Report · Decision Letter 2]

2 Apr 2024

DFMG decreases angiogenesis to uphold plaque stability by inhibiting the TLR4/VEGF pathway in mice

PONE-D-23-29456R2

Dear Dr. Bai,

We’re pleased to inform you that your manuscript has been judged scientifically suitable for publication and will be formally accepted for publication once it meets all outstanding technical requirements.

Kind regards,

Dr. Arvin Haj-Mirzaian

Academic Editor

PLOS ONE

---

## [Editor Report · Acceptance letter]

8 Apr 2024

PONE-D-23-29456R2 

PLOS ONE

Dear Dr. Bai, 

I'm pleased to inform you that your manuscript has been deemed suitable for publication in PLOS ONE. Congratulations! Your manuscript is now being handed over to our production team.

Kind regards, 

on behalf of

Dr. Arvin Haj-Mirzaian 

Academic Editor

PLOS ONE